# PLAN-BASED RELAXED REWARD SHAPING FOR GOAL-DIRECTED TASKS

**Ingmar Schubert[1], Ozgur S. Oguz[2,3], and Marc Toussaint[1,2]**
[1] Learning and Intelligent Systems Group, TU Berlin, Germany
[2] Max Planck Institute for Intelligent Systems, Stuttgart, Germany
[3] Machine Learning and Robotics Lab, University of Stuttgart, Germany
{ingmar.schubert,toussaint}@tu-berlin.de
ozgur.oguz@ipvs.uni-stuttgart.de

## ABSTRACT

In high-dimensional state spaces, the usefulness of Reinforcement Learning (RL) is limited by the problem of exploration. This issue has been addressed using potential-based reward shaping (PB-RS) previously. In the present work, we introduce Final-Volume-Preserving Reward Shaping (FV-RS). FV-RS relaxes the strict optimality guarantees of PB-RS to a guarantee of preserved long-term behavior. Being less restrictive, FV-RS allows for reward shaping functions that are even better suited for improving the sample efficiency of RL algorithms. In particular, we consider settings in which the agent has access to an approximate plan. Here, we use examples of simulated robotic manipulation tasks to demonstrate that plan-based FV-RS can indeed significantly improve the sample efficiency of RL over plan-based PB-RS.

## 1 INTRODUCTION

Reinforcement Learning (RL) provides a general framework for autonomous agents to learn complex behavior, adapt to changing environments, and generalize to unseen tasks and environments with little human interference or engineering effort. However, RL in high-dimensional state spaces generally suffers from a difficult exploration problem, making learning prohibitively slow and sample-inefficient for many real-world tasks with sparse rewards.

A possible strategy to increase the sample efficiency of RL algorithms is reward shaping (Mataric, 1994; Randløv & Alstrøm, 1998), in particular potential-based reward shaping (PB-RS) (Ng et al., 1999). Reward shaping provides a dense reward signal to the RL agent, enabling it to converge faster to the optimal policy. In robotics tasks, approximate domain knowledge is often available and can be used by a planning algorithm to generate approximate plans. Here, the resulting plan can be provided to the RL agent using plan-based reward shaping (Grzes & Kudenko, 2008; Brys et al., 2015). Thus, plan-based reward shaping offers a natural way to combine the efficiency of planning with the flexibility of RL. We analyze the use of plan-based reward shaping for RL. The key novelty is that we theoretically introduce Final-Volume-Preserving Reward Shaping (FV-RS), a superset of PB-RS. Intuitively speaking, FV-RS allows for shaping rewards that convey the information encoded in the shaping reward in a more direct way than PB-RS, since the value of following a policy is not only determined by the shaping reward at the end of the trajectory, but can also depend on all intermediate states.

While FV-RS inevitably relaxes the optimality guarantees provided by PB-RS, we show in the experiments that FV-RS can significantly improve sample efficiency beyond PB-RS, e.g. allowing RL agents to learn simulated 10-dimensional continuous robotic manipulation tasks after ca. 300 rollout episodes. We argue that the strict notion of optimality in PB-RS is not necessary in many robotics applications, while on the other hand relaxing PB-RS to FV-RS facilitates speeding up the learning process. Using FV-RS could be a better trade-off between optimality and sample efficiency in many domains. The contributions of this work are:

- We introduce FV-RS as a new class of reward shaping for RL methods in general.

- We propose to specifically use FV-RS for plan-based reward shaping.

- We show that compared to no RS and plan-based PB-RS, plan-based FV-RS significantly increases the sample efficiency in several robotic manipulation tasks.

## 2 RELATED WORK

### 2.1 SPARSE REWARDS AND REWARD SHAPING

In many real-world RL settings, the agent is only given sparse rewards, exacerbating the exploration problem. There exist several approaches in the literature to overcome this issue. These include mechanisms of intrinsic motivation and curiosity (Barto et al., 2004; Oudeyer et al., 2007; Schembri et al., 2007), which provide the agent with additional intrinsic rewards for events that are novel, salient, or particularly useful for the learning process. In reward optimization (Sorg et al., 2010; Sequeira et al., 2011; 2014), the reward function itself is being optimized to allow for efficient learning.

Similarly, reward shaping (Mataric, 1994; Randløv & Alstrøm, 1998) is a technique to give the agent additional rewards in order to guide it during training. In PB-RS (Ng et al., 1999; Wiewiora, 2003; Wiewiora et al., 2003; Devlin & Kudenko, 2012), this is done in a way that ensures that the resulting optimal policy is the same with and without shaping. Ng et al. (1999) also showed that the reverse statement holds as well; PB-RS is the only type of modification to the reward function that can guarantee such an invariance if no other assumptions about the Markov Decision Process (MDP) are made.

In this work, we introduce Final-Volume-Preserving Reward Shaping (FV-RS), a subclass of reward shaping that is broader than PB-RS and not necessarily potential-based, and therefore is not guaranteed to leave the optimal policy invariant. However, FV-RS still guarantees the invariance of the asymptotic state of the MDP under optimal control. In the experiments section, we show that this relaxed notion of reward shaping allows us to substantially improve the sample efficiency during training.

### 2.2 DEMONSTRATION- AND PLAN-BASED REWARD SHAPING

Learning from Demonstration (LfD) aims at creating a behavioral policy from expert demonstrations. Existing approaches differ considerably in how the demonstration examples are collected and how the policy is derived from this (Argall et al., 2009; Ravichandar et al., 2020). The HAT algorithm (Taylor et al., 2011) introduces an intermediate policy summarization step, in which the demonstrated data is translated into an approximate policy that is then used to bias exploration in a final RL stage. In Hester et al. (2017), the policy is simultaneously trained on expert data and collected data, using a combination of supervised and temporal difference losses. In Salimans & Chen (2018), the RL agent is at the start of each episode reset to a state in the single demonstration. Other approaches (Thomaz et al., 2006; Knox & Stone, 2010) rely on interactive human feedback during the training process.

At the intersection of RL and LfD, reward shaping offers a natural way to include expert demonstrations or plans of the correct behavior into an RL training process. Prior work in this area includes using abstract plan knowledge represented in the form of STRIPS operators to create a potential function for PB-RS (Grzes & Kudenko, 2008; Efthymiadis et al., 2016; Devlin & Kudenko, 2016), which has been applied to the strategy game of Starcraft (Efthymiadis & Kudenko, 2013). Brys et al. (2015) use expert demonstrations to directly construct a Gaussian potential function, and in Suay et al. (2016), this is extended to include multiple demonstrations that are translated into a potential function using Inverse Reinforcement Learning as an intermediate step.

In this work, we use a planned sequence in state-space to construct a shaping function similar to Brys et al. (2015), but in contrast to the aforementioned work, we do not use this shaping function as a potential function for PB-RS. Instead, we use it directly as a reward function for FV-RS. We show that this significantly improves the sample efficiency during training.

## 3 BACKGROUND

### 3.1 MARKOV DECISION PROCESSES AND REINFORCEMENT LEARNING

We consider decision problems that can be described as a discrete-time MDPs (Bellman, 1957) $\langle \mathbb{S}, \mathbb{A}, T, \gamma, R \rangle$. Here, $\mathbb{S}$ is the set of all possible states, and $\mathbb{A}$ is the set of all possible actions. $T : \mathbb{S} \times \mathbb{A} \times \mathbb{S} \to [0, 1]$ describes the dynamics of the system; $T(s'|s, a)$ is the probability (density) of the next state being $s'$, provided that the current state is $s$ and the action taken is $a$. After the transition, the agent is given a reward $R(s, a, s')$. The discount factor $\gamma \in [0, 1)$ trades off immediate and future reward.

The goal of RL is to learn an optimal policy $\pi^* : \mathbb{S}, \mathbb{A} \to [0, 1]$ that maximizes the expected discounted return, i.e.

$$\pi^* = \operatorname*{argmax}_{\pi} \sum_{t=0}^{\infty} \gamma^t \mathbb{E}_{s_{t+1} \sim T(\cdot|s_t, a_t),\, a_t \sim \pi(\cdot|s_t), s_0 \sim p_0} \left[ R(s_t, a_t, s_{t+1}) \right], \tag{1}$$

where $p_0$ is the initial-state distribution, from collected transition and reward data $D = \{(s_i, a_i, s_i')\}_{i=0}^{n}$. The $Q$-function of a given policy $\pi$ is the expected return for choosing action $a_0$ in state $s_0$, and following $\pi$ thereafter, i.e.

$$Q^\pi(s_0, a_0) = \mathbb{E}_{s_{t+1} \sim T(\cdot|s_t, a_t),\, a_t \sim \pi(\cdot|s_t)} \left[ \sum_{t=0}^{\infty} \gamma^t R(s_t, a_t, s_{t+1}) \right] . \tag{2}$$

There exists a range of RL algorithms to solve equation 1. Our reward-shaping approach only modifies the reward and therefore can be combined with any RL algorithm. In this work, we are interested in applications in robotics, where both $\mathbb{S}$ and $\mathbb{A}$ are typically continuous. A popular algorithm in this case is Deep Deterministic Policy Gradient (DDPG) (Lillicrap et al., 2015), which will be used for most of the robotic manipulation examples in this work. An exception to this are the examples in appendix A.4, which use Proximal Policy Optimization (PPO) (Schulman et al., 2017).

### 3.2 POTENTIAL-BASED REWARD SHAPING

In many RL problems, rewards are sparse, making it harder for the RL algorithm to converge to the optimal policy. One possibility to alleviate this problem is to modify the reward $R$ of the original MDP with a shaping reward $F$.

$$\tilde{R}(s, a, s') = R(s, a, s') + F(s, a, s') \tag{3}$$

In general, the optimal policy $\tilde{\pi}^*$ of the MDP $\tilde{M}$ with the shaped reward $\tilde{R}$ is different from the optimal policy $\pi^*$ of the MDP $M$ with the original reward $R$. Ng et al. (1999) showed however that $\tilde{\pi}^* \equiv \pi^*$ if and only if $F(s, a, s')$ is derived from a potential function $\Phi : \mathbb{S} \to \mathbb{R}$:

$$F(s, a, s') = \gamma \Phi(s') - \Phi(s) \tag{4}$$

This proof was extended (Wiewiora et al., 2003) to potential functions of both actions and states, provided that the next action taken, $a'$, is known already. A further generalization to time-dependent potential functions was introduced in Devlin & Kudenko (2012).

## 4 FINAL-VOLUME-PRESERVING REWARD SHAPING

In the following, we introduce the notion of the optimal final volume, which will then allow us to introduce Final-Volume-Preserving Reward Shaping (FV-RS). FV-RS does not guarantee the invariance of the optimal policy like PB-RS does, but instead provides a less restrictive guarantee of preserved long-term behavior.

### 4.1 OPTIMAL FINAL VOLUME

**Definition 4.1.** *Let $M$ be an MDP with state-space $\mathbb{S}$. If the set $\mathbb{G} = \cap_{\mathbb{G}_a \in \mathbb{A}} \mathbb{G}_a$ is non-empty, we call $\mathbb{G}$ the **optimal final volume of M**. Here, $\mathbb{A}$ is the set of non-empty sets $\mathbb{G}_a \subseteq \mathbb{S}$ such that*

$$\exists t_0 > 0 : \quad P_{s_t \sim q_{\pi^*}}(s_t \in \mathbb{G}_a) = 1 \quad \forall t \geq t_0 . \tag{5}$$

Thus, the optimal final volume of an MDP $M$ is the minimal subset of the state space $\mathbb{S}$ of $M$ that the MDP can be found in under optimal control with probability 1 after finite time.

Notice that in order to keep the notation more concise, we use the strict requirement $P_{s_t \sim q_{\pi*}}(s_t \in \mathbb{G}) = 1$ instead of e.g. $P_{s_t \sim q_{\pi*}}(s_t \in \mathbb{G}) \geq 1 - \epsilon$ in definition 4.1. For many stochastic MDPs, the stationary distribution under optimal policy has nonzero density in the entire state space $\mathbb{S}$. This means that strictly speaking, our definition would result in the trivial optimal final volume $\mathbb{G} = \mathbb{S}$ for these MDPs. In practice however, many interesting MDPs are such that we can find a volume $\mathbb{G}$ for which it is so unlikely that the optimal agent will leave it after some finite time has passed that we can readily assume that this probability is 0 for all practical purposes.

## 4.2 COMPATIBLE MDPs AND FINAL-VOLUME-PRESERVING REWARD SHAPING

**Definition 4.2.** *Let $M = \langle \mathbb{S}, \mathbb{A}, T, \gamma, R \rangle$ and $\tilde{M} = \langle \mathbb{S}, \mathbb{A}, T, \tilde{\gamma}, \tilde{R} \rangle$ be MDPs that are identical apart from different reward functions $R$ and $\tilde{R}$, as well as different discount factors $\gamma$ and $\tilde{\gamma}$. Let both $M$ and $\tilde{M}$ have optimal final volumes $\mathbb{G}$ and $\tilde{\mathbb{G}}$, respectively. We call $\tilde{M}$ **a compatible MDP to M** iff $\tilde{\mathbb{G}} \subseteq \mathbb{G}$, and we write $\tilde{M} \subseteq M$. If this is the case, we also call $\tilde{R}$ a **final-volume-preserving** reward shaping function.*

If $\tilde{M} \subseteq M$, this means that although the rewards and discounts $(R, \gamma)$ and $(\tilde{R}, \tilde{\gamma})$ are different and, in contrast to PB-RS, in general result in different optimal policies, they are similar in the sense that after a finite time, the state of $\tilde{M}$ under optimal control will be inside the optimal final volume $\mathbb{G}$ of $M$. In other words; in the long run, behaving optimally in $\tilde{M}$ will have the same "result" as behaving optimally in $M$.

Since PB-RS leaves the optimal policy invariant, $M$ and $\tilde{M}$ are also compatible if $\tilde{M}$ is the PB-RS counterpart to $M$. Thus, the notion of compatibility is less restrictive than the notion of an invariant optimal policy; FV-RS is a superset of PB-RS.

## 4.3 FV-RS

We now proceed to describe specific recipes to find compatible counterparts for MDPs with sparse reward.

**Theorem 4.1.** *Let $M$ be an MDP with state space $\mathbb{S}$ and with the sparse reward function*

$$R(s, a, s') = 1 \quad \text{if } s' \in \mathbb{B}; \quad R(s, a, s') = 0 \quad \text{else}, \tag{6}$$

*where $\mathbb{B} \subseteq \mathbb{S}$ is non-empty. Let $M$ have the optimal final volume $\mathbb{G}$, and let $\mathbb{G} \subseteq \mathbb{B}$. Let $\tilde{R}(s, a, s')$ be a reward function that fulfills*

$$\tilde{R}(s, a, s') = 1 \quad \text{if } s' \in \mathbb{B}; \quad \tilde{R}(s, a, s') \leq \Delta \quad \text{else}. \tag{7}$$

*Then, for every $0 < \Delta < 1$ there exists a discount factor $0 < \tilde{\gamma} < 1$ such that the MDP $\tilde{M}$ corresponding to $\tilde{R}$ and $\tilde{\gamma}$ is compatible with $M$.*

*Proof.* See appendix A.5. ∎

This theorem gives us a worst-case bound on the reward function: Independently of how small the difference $1 - \Delta$ between the reward in $\mathbb{B}$ and elsewhere is, as long as the conditions in equation 7 are fulfilled, we can select a sufficiently large $\tilde{\gamma} < 1$ that renders $\tilde{M}$ a compatible MDP to $M$.

**Corollary 4.1.1.** *This lower bound on $\tilde{\gamma}$ is $\tilde{\gamma} > \Delta^{1/(t_0 - 1)}$.*

*Proof.* Follows directly from the proof of theorem 4.1. ∎

Even if $t_0$ is unknown but finite, $\tilde{M} \subseteq M$ if $\tilde{\gamma}$ is chosen large enough. Note that so far, we have imposed no restrictions on $\tilde{R}(s, a, s')$ for $s' \notin \mathbb{B}$, other than $\tilde{R}(s, a, s') \leq \Delta$. In that sense, the statements above represent an "upper bound on the lower bound of $\tilde{\gamma}$" if we can not assume any additional structure for $\tilde{R}$. However, if we put more restrictions on $\tilde{R}$, this worst-case bound can

be relaxed. As an example, we discuss the case of a reward function that the agent can follow in a step-wise manner in appendix A.6. There it is shown that, depending on the step width $w$, the lower bound $\tilde{\gamma} > \Delta^{1/(t_0-1)}$ can be relaxed to $\tilde{\gamma} > \Delta^{1/(w-1)}$. In the important case that the agent can follow the reward function monotonically (i.e. $w = 1$), this lower bound becomes 0. For more details and a precise definition of the step width $w$, please refer to appendix A.6.

### 4.4 PLAN-BASED FV-RS

We now describe how to construct a plan-based FV-RS shaping reward function from a plan, given as a sequence in state space. The goal is to use the plan to create a continuous reward function that gives dense feedback to the policy to guide it along the plan.

**Theorem 4.2.** *Let $M$ be an MDP with metric state-space $(\mathbb{S}, d)$ with metric $d : \mathbb{S} \times \mathbb{S} \to \mathbb{R}$ and with the reward function*

$$R(s, a, s') = 1 \quad \text{if } s' \in \mathbb{B}; \quad R(s, a, s') = 0 \quad \text{else}, \tag{8}$$

*where $\mathbb{B} \subseteq \mathbb{S}$ is non-empty. Let $M$ have the optimal final volume $\mathbb{G}$, and let $\mathbb{G} \subseteq \mathbb{B}$. From a planned sequence $(p_0, p_1, ..., p_{L-1})$ in $\mathbb{S}$ with $p_{L-1} \in \mathbb{B}$, we can construct the reward function*

$$\tilde{R}(s, a, s') = 1 \quad \text{if } s' \in \mathbb{B}; \quad \tilde{R}(s, a, s') = (1 - \Delta)\frac{k(s') + 1}{L} g(d(p_{k(s')}, s')) \quad \text{else}, \tag{9}$$

*where $k(s') = \operatorname{argmin}_i(d(p_i, s'))$ and $g : \mathbb{R}^{0+} \to (0, 1]$ is strictly monotonically decreasing, where $g(0) = 1$. Then, for every $0 < \Delta < 1$ there exists $0 < \tilde{\gamma} < 1$ such that the MDP $\tilde{M}$ corresponding to $\tilde{R}$ and $\tilde{\gamma}$ is a compatible MDP to $M$.*

*Proof.* Special case of theorem 4.1. ∎

This translates the plan into a continuous reward function that leads towards the target area $\mathbb{B}$. The corresponding MDP $\tilde{M}$ is guaranteed to result in the same optimal final volume as $M$. Thus, we established the machinery to translate any MDP $M$ with the sparse reward function in equation 8 into its plan-based FV-RS counterpart MDP $\tilde{M}$.

## 5 EXPERIMENTS

We demonstrate the efficiency of using FV-RS for plan-based reward shaping using several examples. We start with a simple discrete toy example to illustrate the difference between PB-RS and FV-RS that is not potential-based. We then compare PB-RS and FV-RS using a realistic 10-dimensional simulated robotic pushing example. This is one of several robotic pushing and pick-and-place examples we investigated. All results can be found in appendix A.2. Videos of all manipulation examples can be found in the supplementary material.

### 5.1 DISCRETE TOY EXAMPLE

We start off considering a simple discrete example as shown in figure 1a. The agent starts in the middle and can choose between the actions $\mathbb{A} = \{\texttt{left}, \texttt{right}, \texttt{grab}\}$, where $\texttt{grab}$ does not change the state, but allows the agent to grab the flag provided that it is already at the position of the flag. The reward function is shown in figure 1b. Without shaping, reward 1 is only given if the flag is captured. We also define the potential-based reward function

$$\tilde{R}_{\text{PB}}(s, a, s') = R(s, a, s') + \tilde{\gamma}\Phi(s') - \Phi(s); \qquad \Phi(s) = \begin{cases} 0.5 & \text{if } s = \texttt{finish} \\ 0 & \text{else} \end{cases}, \tag{10}$$

where the shaping potential $\Phi(s)$ assigns value to being at the correct position, even if the flag is not captured. Similarly, we define the reward function

$$\tilde{R}_{\text{FV}}(s, a, s') = \begin{cases} 1 & \text{if } s' = \texttt{finish} \text{ and } a = \texttt{grab} \\ 0.5 & \text{if } s' = \texttt{finish} \text{ and } a \neq \texttt{grab} \\ 0 & \text{else} \end{cases}, \tag{11}$$

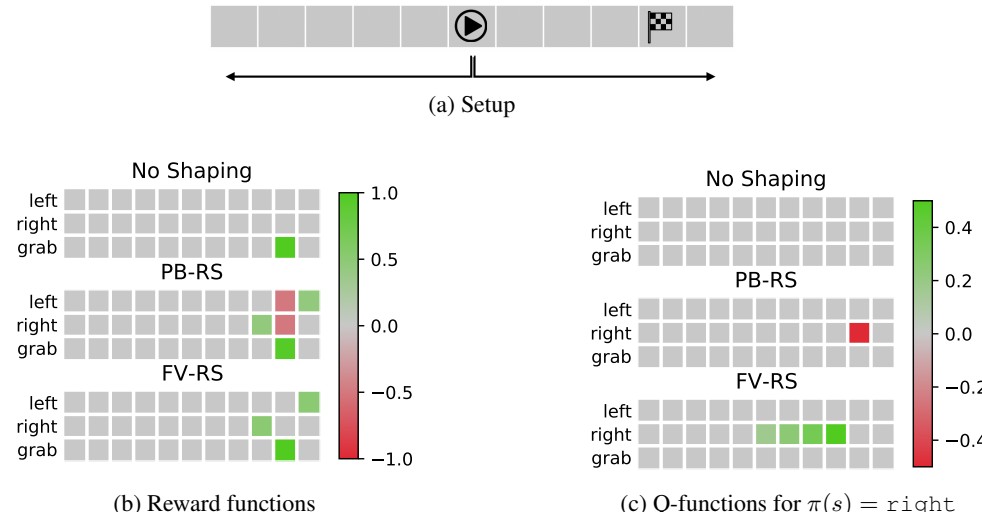

(a) Setup

(b) Reward functions

(c) Q-functions for $\pi(s) = \texttt{right}$

Figure 1: The toy example described in section 5.1. (a) The agent starts in the middle, and can move to the right and to the left. The goal is to grab the flag when at the goal position. The agent collects data during two episodes indicated by the arrows. (b) The reward functions without shaping, with PB-RS, and with FV-RS. (c) The resulting Q-functions for policy $\pi(s) = \texttt{right}$, based on the data from the two episodes.

which is not potential-based but final-volume-preserving for the discount factor $\gamma = \tilde{\gamma} = 0.7$ of the original MDP. The agent collects data during 2 rollout episodes starting in the middle, during one of which it always chooses to move $\texttt{right}$ until the end of the episode, and another episode during which the agent always moves $\texttt{left}$. Assume that this data is used in an actor-critic setup, where the actor policy is not converged yet and therefore always chooses the action $\texttt{right}$. For this given policy and given data, the resulting $Q$-functions are shown in figure 1c. Without reward shaping, there is no reward signal collected during the rollouts, and therefore naturally, the $Q$-function contains no information. With PB-RS, the reward for moving to the correct position is exactly canceled out by the discounted penalty for moving away from it again. As a result, the values of the $Q$-function at the starting position contain no information about which action is preferable after these two training rollouts. With FV-RS, the shaping reward is not canceled out and propagated all the way through to the origin. In this case, the $Q$-function provides the agent with the information that moving to the right is preferable if it finds itself at the starting position.

With PB-RS, the shaped return that is assigned to following a certain policy only depends on the discounted difference of the rewards at the final step and initial step of the recorded episode (Grzes, 2017). For non potential-based shaping reward functions like $\tilde{R}_{\text{FV}}$ however, the shaped return depends on all intermediate rewards on the way there as well. In that sense, FV-RS allows for shaped reward functions that propagate the reward information of intermediate states faster than PB-RS reward functions.

To use a physical analogy, PB-RS is analogous to a conservative force field, in which the potential energy of a particle only depends on its current position. FV-RS that is not potential-based is like a non-conservative (e.g. friction) force, for which the energy of a particle is not only a function of its position, but a function of the entire past trajectory.

## 5.2 ROBOTIC PUSHING TASK

In this example, we test the efficiency of FV-RS in a simulated robotic pushing task with realistic physics. The task is very similar to the $\texttt{FetchPush-v1}$ environment in the OpenAI Gym (Brockman et al., 2016), but is implemented using open-source software. Specifically, we use the NVIDIA PhysX engine (phy, 2020) to simulate a box of size $0.4 \times 0.4 \times 0.2$ lying on a table of size $3 \times 3$, as shown in figure 2a.

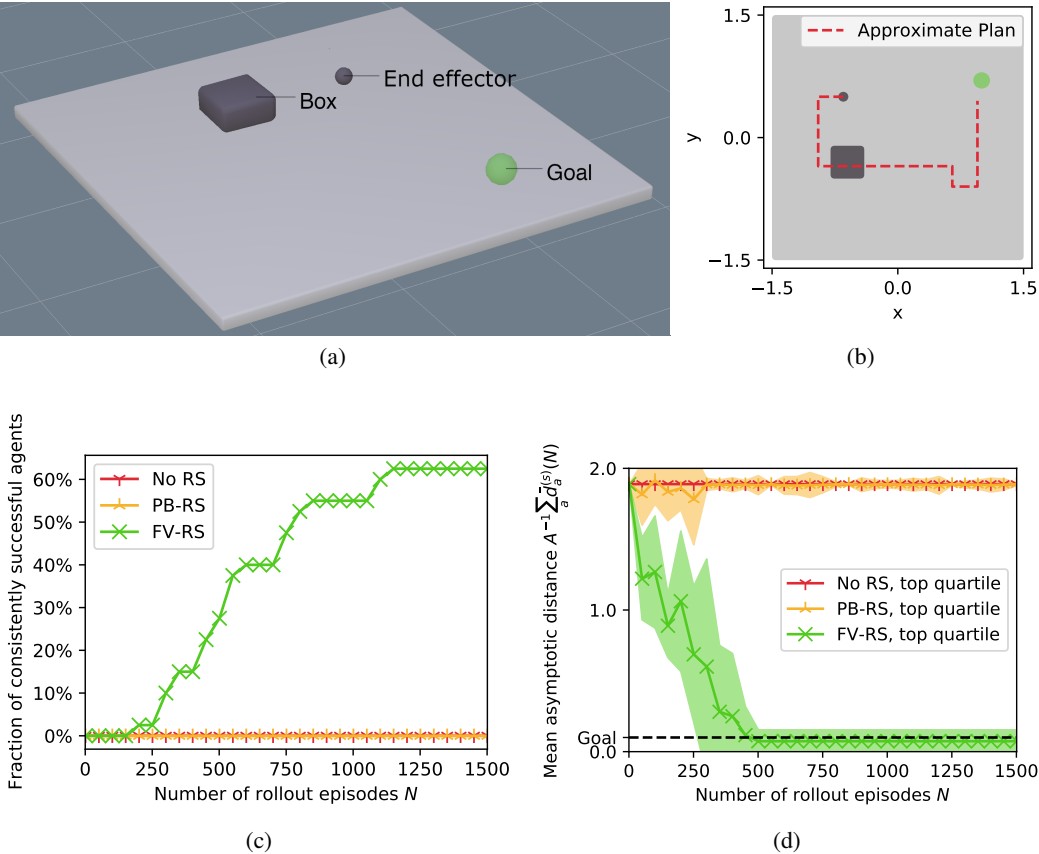

Figure 2: Robotic pushing task. (a) A box of size $0.4 \times 0.4 \times 0.2$ (dark gray) lying on a table of size $3 \times 3$ (light gray) is supposed to be pushed to the green position by the spherical end effector (dark gray). (b) Top view of the table with 2-D projection of the planned 10-D trajectory (x and y coordinates of the end effector). The planned trajectory is not executable in the noisy environment. (c) With FV-RS, some agents are consistently successful after around 300 training rollout episodes. Without reward shaping or with PB-RS, none of the agents were able to consistently reach the goal in this experiment. (d) Training progress of the top quartile of agents out of each category.

The agent controls the 3-dimensional velocity of a spherical end effector of radius $0.06$. The maximum velocity in any direction is $0.1$ per time step. The actual movement of the end effector is noisy; in $10\%$ of the time steps, the end effector moves randomly. Using quaternion representation, the state space $\mathbb{S}$ has $7 + 3 = 10$ degrees of freedom. The goal is to push the box within a distance of $0.1$ from the goal position shown in green, with any orientation. Let $\mathbb{B}$ be the set of states that fulfill this requirement. The unshaped reward $R(s, a, s')$ is 1 if $s' \in \mathbb{B}$ and is 0 elsewhere.

The plan $(p_0, p_1, ..., p_{L-1})$ is given as a sequence of states, as shown in figure 2b. The planned sequence has been created manually using a noise-free model, and therefore can not be followed directly in the realistic simulation. Instead, it is used to create a plan-based shaped reward from it. Following the suggestion of theorem 4.2, we specifically compare the reward functions

$$
\tilde{R}_{\text{FV}}(s, a, s') = \begin{cases} 1 & \text{if } s' \in \mathbb{B} \\ \Phi_{\text{FV}}(s') & \text{else} \end{cases} \quad ; \quad \tilde{R}_{\text{PB}}(s, a, s') = R(s, a, s') + (\tilde{\gamma}\Phi_{\text{PB}}(s') - \Phi_{\text{PB}}(s)).
$$

$$(12)$$

If $\Phi_{\text{FV}}$ is chosen suitably following theorem 4.2, $\tilde{R}_{\text{FV}}$ is final-volume-preserving with respect to reward function $R$ of the original MDP. The potential-based function $\tilde{R}_{\text{PB}}$ acts as the baseline. Notice that $\Phi_{\text{FV}}$ and $\Phi_{\text{PB}}$ are chosen independently in order to facilitate a fair comparison of FV-RS and PB-RS. We discuss and compare different choices both for $\Phi_{\text{FV}}$ and $\Phi_{\text{PB}}$ in appendix A.1. There

we compare different gaussian plan-based functions, similar to the ones used in Brys et al. (2015), as well as a more simple function that only depends on the distance of the box to the target. Our analysis reveals that using the gaussian plan-based function

$$\Phi_{\text{FV}}(s) = \frac{1}{2} \frac{k(s)+1}{L} \exp\left(-\frac{d^2(s, p_{k(s)})}{2\sigma^2}\right) \tag{13}$$

with $\sigma = 0.2$ results in a robust performance for FV-RS. Here, $k(s) = \text{argmin}_i(d(p_i, s))$, and $d(\cdot, \cdot)$ is the euclidean distance ignoring the coordinates corresponding to the orientation of the box. The value of this function increases if the agent moves further along the planned path or closer towards the planned path. Similarly, our analysis also shows that using $\Phi_{\text{PB}}(s) = K\Phi_{\text{FV}}(s)$ with $K = 50$ results in a robust performance for PB-RS. We would like to emphasize that we did not impose that $\Phi_{\text{PB}}(s)$ is a multiple of $\Phi_{\text{FV}}(s)$, rather our analysis revealed that this is the most suitable choice. For more details, please refer to appendix A.1.

We apply DDPG (Lillicrap et al., 2015) for the examples presented throughout this paper, with the exception of the example presented in appendix A.4 using PPO (Schulman et al., 2017). We use $\gamma = \tilde{\gamma} = 0.9$ as discount factor. The agent collects data in rollout episodes of random length sampled uniformly between 0 and 300. Each of these episodes starts at the same initial position indicated in figure 2a; we do not assume that the system can be reset to arbitrary states. The exploration policy acts $\epsilon$-greedily with respect to the current actor, where $\epsilon = 0.2$. After every rollout, actor and critic are updated using the replay buffer. Both actor and critic are implemented as neural networks in Tensorflow (Abadi et al., 2016). Implementation details can be found in the supplementary code material.

The data reported in figure 2 is obtained as follows: For each of the shaping strategies $s \in \{\text{No RS}, \text{PB-RS}, \text{FV-RS}\}$, multiple agents $a \in \{1, ..., A\}$ are trained independently from scratch. After $N$ rollout episodes, multiple test episodes $m \in \{1, ..., M\}$ are run for 300 time steps or until the goal is reached. $d_{am}^{(s)}(N)$ is the distance of the box to the goal at the end of such a test episode. We classify an agent as being consistently successful once $\bar{d}_a^{(s)} = 1/M \sum_m d_{am}^{(s)}(N) < 0.1$. We emphasize that this is a rather strict criterion. Over $M$ test rollouts, a consistently successful agent must achieve an *average* asymptotic distance to the goal that is within the margin of 0.1, meaning that essentially all test rollouts have to be within the margin. We use $A = 40$ and $M = 30$ in all experiments reported. The training of an agent $a$ is halted once it is consistently successful.

We report data from all agents in figure 2c and focus on the best performing quartile of agents out of each category in figure 2d. The performance of agents is ranked by how quickly they become consistently successful, or by how close to the goal they get in case they never become consistently successful. FV-RS helps to learn the task with significantly higher sample efficiency than PB-RS and without reward shaping. Some agents (see figure 2c) are consistently successful after ca. 300 rollout episodes (corresponding to ca. $45\,000$ MDP transitions) using FV-RS. In contrast, none of the agents using PB-RS or no reward shaping were able to consistently reach the goal after 1500 training rollouts in this example.

Apart from this pushing example, we also investigated other robotic pushing or pick-and-place settings in appendix A.2. Furthermore, we compare FV-RS and PB-RS using PPO instead of DDPG in appendix A.4. While DDPG is off-policy and deterministic, PPO is on-policy and stochastic. Consistently across all experiments, we find a significant advantage of using FV-RS over PB-RS, which qualitatively confirms the findings discussed here. Videos of this and other manipulation examples can also be found in the supplementary material.

## 6 DISCUSSION AND CONCLUSIONS

We introduced FV-RS, a subclass of reward shaping that relaxes the invariance guarantee of PB-RS to a guarantee of preserved long-time behavior. We illustrated that since FV-RS is not necessarily potential-based, it can convey the information provided by the shaping reward in a more direct way than PB-RS. Using FV-RS, the value of following a policy does not necessarily depend on the discounted final reward only, but can also depend on all intermediate rewards. Thus, FV-RS is analogous to a non-conservative force (e.g. friction), and PB-RS to a conservative force (e.g. gravity).

We proposed to use FV-RS in order to increase the sample efficiency of plan-informed RL algorithms. We demonstrated in the experiments that, compared to plan-based PB-RS, plan-based FV-RS can increase the sample efficiency of RL significantly. This observation is consistent across multiple manipulation examples and RL algorithms. In all manipulation examples tested, there were agents that were consistently successful after ca. 300 rollout episodes using FV-RS and DDPG.

An important limitation of plan-based reward shaping in general arises if the plan used for the shaping is globally wrong. In such a case, the shaped reward would be of no use for the exploration and potentially misleading. Here, combining FV-RS with other exploration strategies could possibly remedy this limitation. Another limitation of the presented method is that it is currently formulated for a goal-driven sparse-reward setting. We restricted the present paper to the discussion of this class of MDPs since it is often encountered e.g. in robotic manipulation tasks. The direct application of RL still poses a major obstacle here, making reward shaping an important tool.

Since reward shaping only modifies the reward itself, our approach makes no other assumptions about the RL algorithm used. It is orthogonal to and can be combined with other techniques used in RL, including popular methods for sparse-reward settings like Hindsight Experience Replay (Andrychowicz et al., 2017). The same holds true for methods in LfD that also exploit task information provided in the form of an approximate demonstration or plan. Plan-based FV-RS only relies on a relatively simple representation of the approximate plan in the form of a sequence in state space. In addition, our method does not require the system to be reset to arbitrary states, as it is the case e.g. in Salimans & Chen (2018).

Both these aspects make it a practical choice for real-world robotic applications. In many robotic manipulation settings, an approximate planner for the task is available, but the resulting plan can not be directly executed on the real robot. Plan-based FV-RS could facilitate this *sim-2-real transfer* in an efficient and principled way using RL. Trading strict optimality guarantees for increased sample efficiency (while still guaranteeing preserved long-time behavior) could be beneficial in these cases.

### ACKNOWLEDGMENTS

We thank the anonymous reviewers for their insightful suggestions and comments. We thank the MPI for Intelligent Systems for the Max Planck Fellowship funding.

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

## A  APPENDIX

Appendix A.1 compares different choices of shaping functions used for PB-RS and for FV-RS. The particular choice that was used for the experiments in this paper is also discussed and justified there. Appendix A.2 contains additional robotic manipulation examples. These are instances of both the simulated robotic pushing task presented in section 5.2 and a simulated robotic pick-and-place task presented in appendix A.3. Appendix A.4 contains robotic pushing examples using PPO as RL algorithm instead of DDPG. Appendix A.5 contains the proof of theorem 4.1. In appendix A.6, a special case of theorem 4.1 is discussed, additionally assuming a step-wise reward function that the agent can follow monotonically.

### A.1  EXPERIMENTS ON THE CHOICE OF SHAPING FUNCTIONS USED FOR PB-RS AND FV-RS

We assume that the plan $(p_0, p_1, ..., p_{L-1})$ is given as a sequence of states. The functions $\tilde{R}_{\text{FV}}$ and $\tilde{R}_{\text{PB}}$, and therefore $\Phi_{\text{FV}}$ and $\Phi_{\text{PB}}$, should provide a reward that is higher the closer the state of the environment is to the plan, and the further advanced it is along the plan towards the goal. Furthermore, $\Phi_{\text{FV}}(s) < 1 \; \forall s \in \mathbb{S}$ has to be fulfilled in order to ensure that $\tilde{R}_{\text{FV}}$ fulfills the requirements of theorem 4.2. A canonical choice that satisfies these constraints are the Gaussian functions

$$\Phi_{\text{FV}}^{\text{Gauss}}(s; \sigma) = \frac{1}{2} \frac{k(s) + 1}{L} \exp\left(-\frac{d^2(s, p_{k(s)})}{2\sigma^2}\right); \quad \Phi_{\text{PB}}^{\text{Gauss}}(s; \sigma) = K \Phi_{\text{FV}}^{\text{Gauss}}(s; \sigma) \quad (14)$$

where $k(s) = \arg\min_i(d(p_i, s))$, $d(\cdot, \cdot)$ is the euclidean distance ignoring the coordinates corresponding to the orientation of the box, and $K = 50$ is a factor that is applied in order to render $\tilde{R}_{\text{FV}}$ and $\tilde{R}_{\text{PB}}$ comparable in scale. $\Phi_{\text{PB}}^{\text{Gauss}}$ is similar to the shaping potential that was used for potential-based reward shaping from demonstration in Brys et al. (2015).

As a simpler alternative, we also compare against the negative distance functions

$$\Phi_{\text{FV}}^{\text{Dist}}(s) = -d_{\text{Box}}(s, p_{L-1}); \quad \Phi_{\text{PB}}^{\text{Dist}}(s) = K \Phi_{\text{FV}}^{\text{Dist}}(s) \quad , \quad (15)$$

where $d_{\text{Box}}(\cdot, p_{L-1})$ denotes the distance of the box the goal located at $p_{L-1}$. These functions do not take the planned sequence into account, but only the intended final state, i.e. the goal. It is easy to verify that using $\Phi_{\text{FV}}^{\text{Dist}}$ results in a final-volume-preserving reward function.

We compare the performance of FV-RS agents using $\Phi_{\text{FV}}^{\text{Dist}}$ or $\Phi_{\text{FV}}^{\text{Gauss}}(\cdot, \sigma)$ for different $\sigma$ against PB-RS agents using $\Phi_{\text{PB}}^{\text{Dist}}$ or $\Phi_{\text{PB}}^{\text{Gauss}}(\cdot, \sigma)$ for different $\sigma$. We use two different robotic pushing examples for this comparison. These are instances of the pushing example described in section 5.2, but with different initial positions and different goal positions. We measure the asymptotic distance over different agents and rollouts as detailed in section 5.2. The results are presented in figure 3.

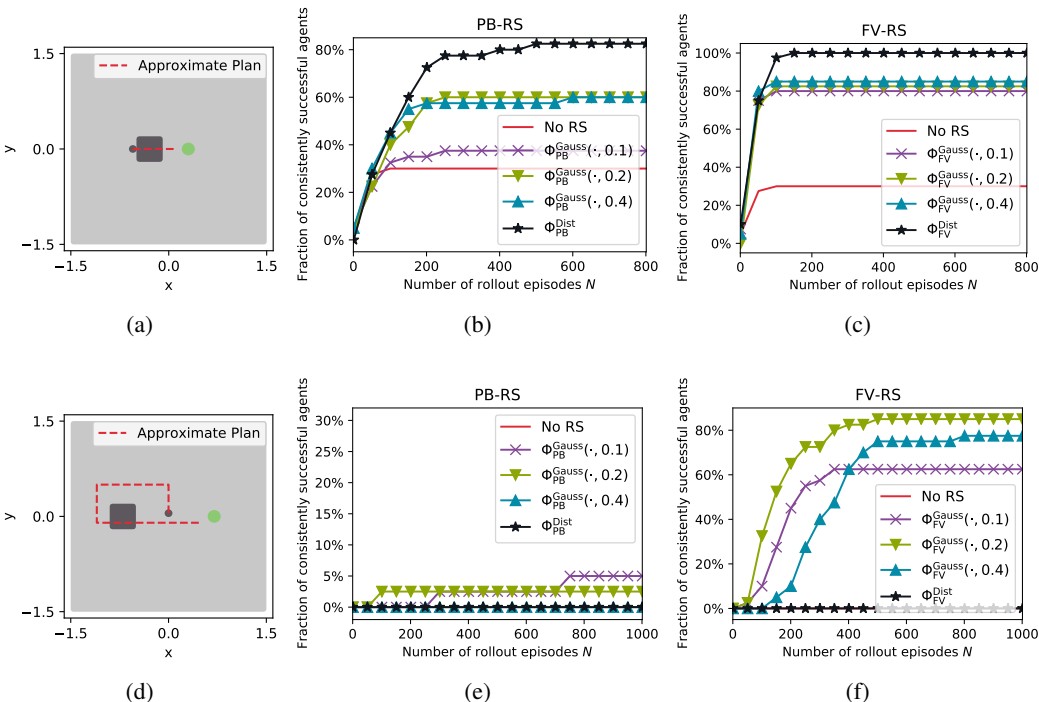

Figure 3: Comparison of the performance of PB-RS and FV-RS when using different reward shaping functions as defined in equation 14 and equation 15. (a) The first row shows a very simple example. (b) In this example, using $\Phi_{\text{PB}}^{\text{Dist}}$ shows the best result with PB-RS, followed by $\Phi_{\text{PB}}^{\text{Gauss}}(\cdot, 0.2)$ or $\Phi_{\text{PB}}^{\text{Gauss}}(\cdot, 0.4)$. (c) Similarly, $\Phi_{\text{FV}}^{\text{Dist}}$ shows the best result with FV-RS, followed by $\Phi_{\text{FV}}^{\text{Gauss}}$, which seems to be robust against different choices of $\sigma$. (d) The second row shows a more difficult example, in which the end effector has to move around the box before initiating the push. (e) Here, using the purely distance-based potential function $\Phi_{\text{PB}}^{\text{Dist}}$ is not successful, and the best result for PB-RS is achieved with $\Phi_{\text{FV}}^{\text{Gauss}}(\cdot, 0.2)$ or $\Phi_{\text{FV}}^{\text{Gauss}}(\cdot, 0.1)$. (f) With FV-RS, a similar result is obtained, although $\Phi_{\text{FV}}^{\text{Gauss}}(\cdot, 0.2)$ performs significantly better here.

Interestingly, $\Phi_{\text{PB}}^{\text{Dist}}$ and $\Phi_{\text{FV}}^{\text{Dist}}$ perform best in the simple example. However, using these functions for reward shaping does not scale well to the more difficult example. This can be ascribed to the missing path information when using purely distance-based information for reward shaping. On the other hand, using $\Phi_{\text{PB}}^{\text{Gauss}}(\cdot, 0.2)$ for PB-RS seems to be the second-best choice for the simple example, and one of the best for the more difficult one. The same holds true for using $\Phi_{\text{FV}}^{\text{Gauss}}(\cdot, 0.2)$ for FV-RS. Since the manipulation problems we are interested in in this paper resemble the more difficult example more than the simpler one, these experiments suggest that using $\Phi_{\text{FV}}^{\text{Gauss}}(\cdot, 0.2)$ for FV-RS is the best and most robust choice. We refer to this function as $\Phi_{\text{FV}}$ throughout the rest of the paper. Furthermore, using $\Phi_{\text{PB}}^{\text{Gauss}}(\cdot, 0.2)$ for the baseline method PB-RS allows for a fair comparison, since this choice has shown the best performance on the more difficult problem using FV-RS. We refer to this function as $\Phi_{\text{PB}}$ throughout the rest of the paper.

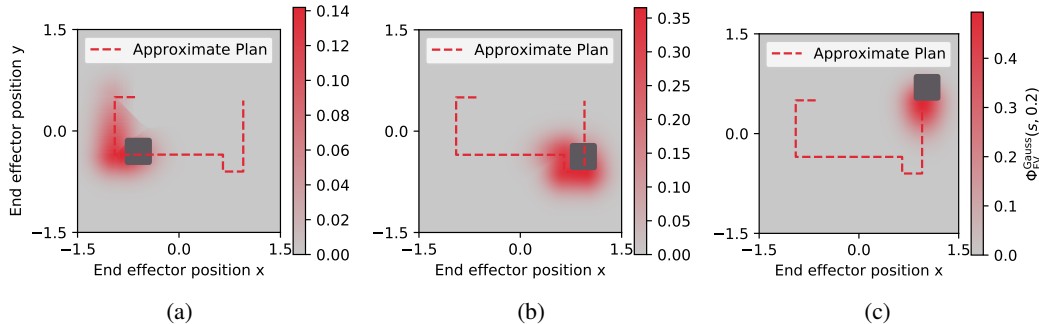

Figure 4: Three different slices of the function $\Phi_{\text{FV}}^{\text{Gauss}}(\cdot, 0.2)$ defined in equation 14, using the example discussed in section 5.2 The state space is 10-dimensional; the figures show the function values for fixed 7-dimensional box position and orientation (indicated by the dark grey boxes) and fixed $z$ position of the end effector which is $z = 0.14$ for all three figures.

On a different note, we observe that independently of the shaping function used, the performance of PB-RS decreases faster when moving to the more difficult example than the performance of FV-RS.

For the example discussed in section 5.2, plots of the function $\Phi_{\text{FV}}^{\text{Gauss}}(\cdot, 0.2)$ are shown in figure 4. $\Phi_{\text{FV}}^{\text{Gauss}}(\cdot, 0.2)$ is larger the closer $s$ is to the approximate plan $(p_0, p_1, ..., p_{L-1})$ given as a sequence of states, and the further advanced along the plan the state $p_i$ closest to $s$ is (compare e.g. the scale of figure 4a to figure 4c). Using the approximate plan, the reward shaping function $\tilde{R}_{\text{FV}}$ that is constructed using $\Phi_{\text{FV}}^{\text{Gauss}}(\cdot, 0.2)$ provides a signal to the RL agent that helps guiding it towards the goal.

## A.2 ADDITIONAL ROBOTIC MANIPULATION EXAMPLES

The robotic manipulation examples presented in the following are instances of the simulated robotic pushing task presented in section 5.2 and a simulated robotic pick-and-place task presented in appendix A.3. The results are summarized in figure 5 and figure 6, respectively.

Qualitatively, the findings discussed in section 5.2 are confirmed across all examples. In both the pushing and the pick-and-place setting, FV-RS helps to learn the manipulation task with significantly higher sample efficiency than PB-RS. In all examples, there were agents that learned to consistently fulfill the task after ca. 300 rollout episodes using FV-RS, which corresponds to $45\,000$ MDP transitions.

The first pushing example of figure 5 can be seen as an easier version of the second example. Interestingly, no agent is able to consistently succeed at the harder task with PB-RS, while one agents out of $A = 40$ is able to consistently succeed at the easier one with PB-RS. In contrast to that, the majority of agents learns to consistently fulfill both tasks after ca. 300 training rollouts in both examples using FV-RS.

A video of all experiments is included in the supplementary material.

## A.3 ROBOTIC PICK-AND-PLACE TASK

For the robotic pick-and-place task, we use the NVIDIA PhysX engine (phy, 2020) to simulate a disk of radius $0.2$ lying on a table of size $3 \times 3$, as shown in figure 7a.

We assume that the grasp is automatically established if the end effector touches the disk in the red area. Action and state space are the same as in section 5.2, apart from the information whether a grasp has been established or not, which is appended to the state of the system (binary signal). Just as in section 5.2, the movement of the end effector is noisy. The goal is to place the disk within a distance of $0.1$ from the goal position, shown in green in in figure 7a. Let $\mathbb{B}$ be the set of states that fulfill this requirement. The unshaped reward $R(s, a, s')$ is $1$ if $s' \in \mathbb{B}$ and is $0$ elsewhere. We choose $\gamma = \tilde{\gamma} = 0.9$

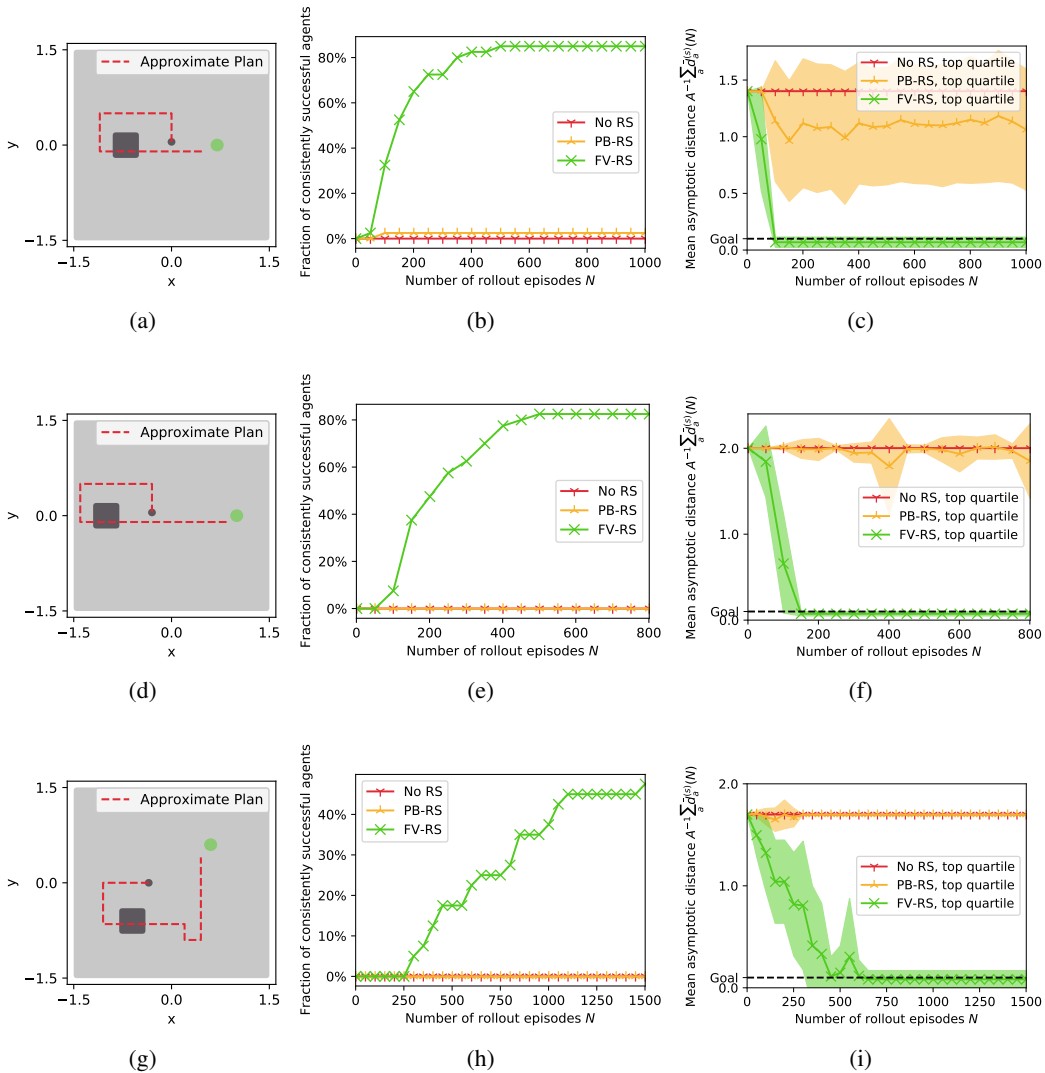

Figure 5: Additional pushing examples. Each row corresponds to a different example. With FV-RS, some agents are consistently successful after ca. 300 rollout episodes in all cases. With PB-RS, there is one agents in the first example that learns to be consistently successful within the 1000 rollout episodes the experiment was run for. Across all examples, the training progress of the top quartile of agents is significantly faster with FV-RS than with PB-RS or without RS. Different evaluations from the first example were also shown in figure 3

Again, the plan (see figure 7b) is given as a sequence of states $(p_0, p_1, ..., p_{L-1})$, and we construct the potential-based shaped reward $\tilde{R}_{PB}$ and the final-volume-preserving reward $\tilde{R}_{FV}$ analogously to equation 12 and equation 13. This is motivated by the analysis done in appendix A.1, combined with the observation that the scale and complexity pick-and-place example is comparable to the scale and complexity of the pushing example. In contrast to equation 13 however, the euclidean distance $d(\cdot, \cdot)$ also takes into account the binary information on whether a grasp has been established or not. For PB-RS, we scale the potential function by $K = 30$. This is done in order to ensure that the reward functions of PB-RS and FV-RS are of similar scale. The results for 2 different pick-and-place examples are discussed in appendix A.2, specifically in figure 6. Here, we measure the asymptotic distance over different agents and rollouts as detailed in section 5.2.

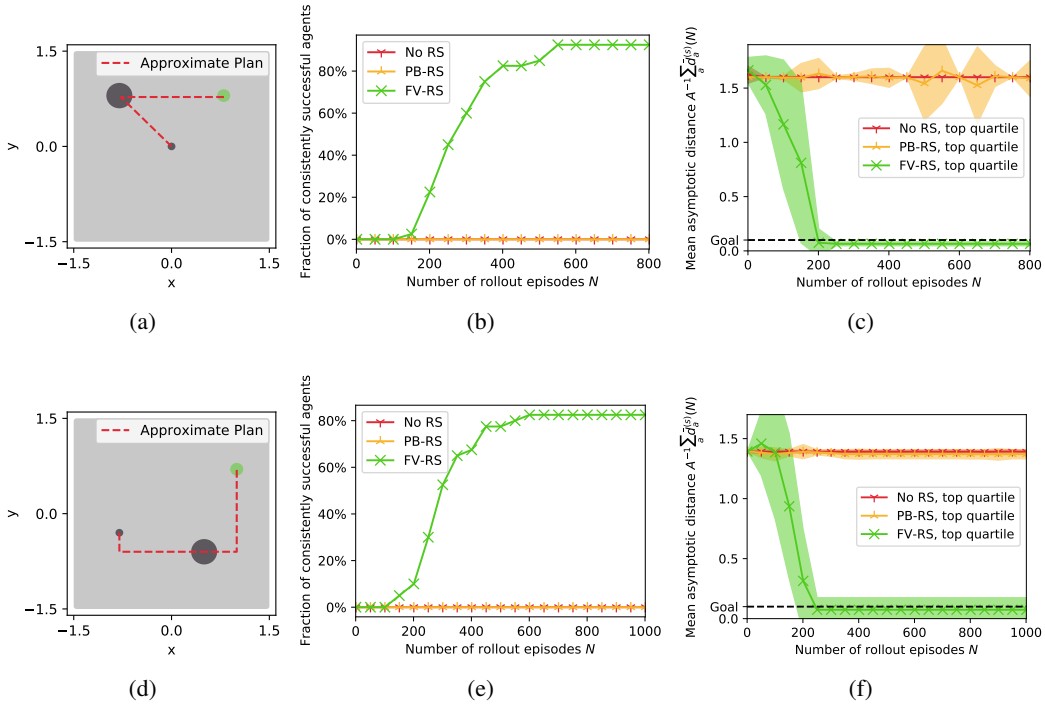

Figure 6: Pick-and-place examples. Each row corresponds to a different example. With FV-RS, some agents are consistently successful after ca. 300 rollout episodes in both cases. With PB-RS or without reward shaping, none of the agents were able to consistently reach the goal in these experiments. The training progress of the top quartile of agents is significantly faster with FV-RS than with PB-RS or without reward shaping.

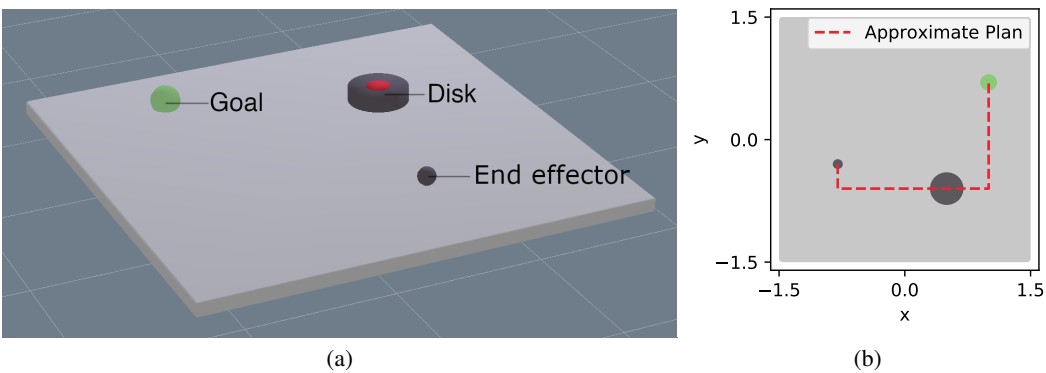

Figure 7: Robotic pick-and-place task. (a) A disk of radius 0.2 (dark gray) lying on a table of size $3 \times 3$ (light gray) is supposed to be picked up in the red area and placed at the green position by the spherical end effector (dark gray). (b) Top view of the table with 2-D projection of a planned trajectory (x and y coordinates of the end effector). The planned trajectory is not executable in the noisy environment.

## A.4   ROBOTIC PUSHING EXAMPLES USING PPO

We repeat the experiments presented in appendix A.1 using PPO instead of DDPG as RL algorithm. In DDPG, a deterministic policy is learned from off-policy data. In contrast to this, PPO is an on-policy algorithm for learning a stochastic policy.

We use an open-source implementation from Barhate (2020). The actor network outputs mean values between $-0.1$ and $0.1$ for the Gaussian policy, while the standard deviation of the policy output is fixed to $0.015$. We use the clipping parameter $\epsilon = 0.1$ (see Schulman et al. (2017)). For more implementation details, please also refer to the supplementary material.

Since the environment is unchanged, we use the shaping functions defined in equation 12 and equation 13. The results are presented in figure 8. The relative advantage of using FV-RS over PB-RS

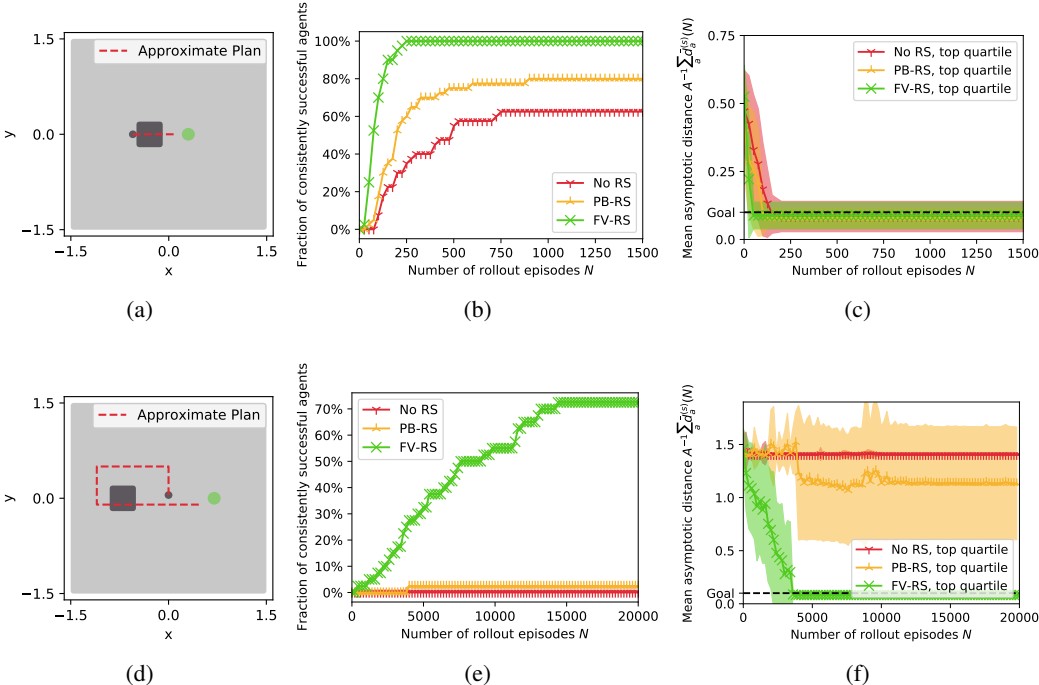

Figure 8: Pushing experiments using PPO instead of DDPG as the RL algorithm. (a, b, c) On the simpler example, FV-RS only slightly outperforms PB-RS. (d, e, f) This difference increases however on the more difficult example. This relative advantage of using FV-RS over PB-RS using PPO is very similar to the one reported in figure 3 for DDPG.

with PPO is very similar to the one reported in figure 3, where results for the same examples using DDPG are reported. This relative advantage is relatively small on the simpler example, but increases drastically for the more difficult example.

As a side note, the absolute number of samples needed by an agent to consistently learn the task is typically much higher using the on-policy PPO algorithm. This is especially true for the more difficult example. Another interesting observation is that without any reward shaping (No RS), the stochastic PPO policy performs better than the deterministic DDPG policy in figure 3. This is however not the case in the more difficult example.

## A.5 Proof of Theorem 4.1

We adopt notation from section 4. The largest reward $1$ can only be collected if $s'$ is within $\mathbb{B}$. Since $M$ has the optimal final volume $\mathbb{G} \subseteq \mathbb{B}$ and $M$ and $\tilde{M}$ have the same dynamics $T$, there exists a policy (the optimal policy of $M$) to drive the state of $\tilde{M}$ to $\mathbb{G} \subseteq \mathbb{B}$ in finite time $t_0$. The only question that remains is whether it is suboptimal for a policy to drive the state of $\tilde{M}$ to $\mathbb{G} \subseteq \mathbb{B}$ if this takes many steps, and whether it might be optimal to stay at a local maximum of $\tilde{R}(s, a, s')$.

The maximum return that can be collected outside of $\mathbb{B}$ is $\Delta/(1 - \tilde{\gamma})$. On the other hand, directly navigating to $\mathbb{G} \subseteq \mathbb{B}$ returns at least $\tilde{\gamma}^{t_0 - 1}/(1 - \tilde{\gamma})$. For every $0 < \Delta < 1$, we can therefore find a $0 < \tilde{\gamma} < 1$ such that directly navigating to $\mathbb{G} \subseteq \mathbb{B}$ is favorable under reward $\tilde{R}(s, a, s')$ and discount

$\tilde{\gamma}$ as well. Once the state is within $\mathbb{G}$, the optimal policy of $M$ can be applied to keep the state within $\mathbb{G}$. Again, this is feasible since $M$ has the optimal final volume $\mathbb{G}$. Thus, $\tilde{M} \subseteq M$.

## A.6 STEP-WISE REWARD FUNCTION

We adopt notation from section 4. We discuss a possible relaxation of the lower bound on $\tilde{\gamma}$ given in corollary 4.1.1 in case more assumptions are made about $\tilde{R}$. Specifically, we consider the case of a step-wise reward function that the agent can follow monotonically.

**Definition A.1.** *Let a reward function $\tilde{R}$ fulfill*

$$\forall s, a, s' \in \mathbb{B}: \quad R(s, a, s') = 1 \tag{16}$$

$$\forall s, a, s' \notin \mathbb{B}: \quad R(s, a, s') \in \{\Delta, \Delta^2, \dots\} \quad, \tag{17}$$

*where $\mathbb{B} \subseteq \mathbb{S}$ is non-empty. For $w \in \mathbb{N} \setminus \{0\}$, we call an MDP $M$ **monotonically** $(w, \Delta)$ **controllable with respect to** $\tilde{R}$, iff there exists a policy $\pi$ and $0 < \Delta < 1$ such that*

$$\forall s_0 \notin \mathbb{B}, \quad \forall k \geq w: \quad P(\Delta \cdot R(s_k, a_k, s_{k+1}) \geq R(s_0, a_0, s_1)) = 1 \quad. \tag{18}$$

*where actions and states follow the distribution*

$$q_\pi(a_0, s_1, a_1, \dots | s_0) = \prod_{i=0}^{\infty} P(s_{t+1} | s_t, a_t) \pi(a_t | s_t) \quad. \tag{19}$$

*We call $w$ $M$'s **step width** and we call $\Delta$ $M$'s **step height**.*

In words, we additionally require that given the dynamics of $M$, the reward function must be defined in a way that guarantees that the agent can advance to the next "step" within $w$ time steps. We acknowledge that constructing such a reward function might not always be feasible, but it allows us to illustrate the relationship between the lower bound on $\tilde{\gamma}$ and the step width $w$. The following theorem states this relationship.

**Theorem A.1.** *Let $M$ be an MDP with metric state-space $(\mathbb{S}, d)$ and with the sparse reward function*

$$R(s, a, s') = \begin{cases} 1 & \text{if } s' \in \mathbb{B} \\ 0 & \text{else} \end{cases}, \tag{20}$$

*where $\mathbb{B} \subseteq \mathbb{S}$ is non-empty. Let $M$ have the optimal final volume $\mathbb{G} \subseteq \mathbb{B}$. Let $\tilde{R}(s, a, s')$ be a reward function that fulfills*

$$\forall s, a, s' \in \mathbb{B}: \quad R(s, a, s') = 1 \tag{21}$$

$$\forall s, a, s' \notin \mathbb{B}: \quad R(s, a, s') \in \{\Delta, \Delta^2, \dots\} \quad, \tag{22}$$

*where $0 < \Delta < 1$. Additionally, let $M$ be monotonically $(w, \Delta)$ controllable with respect to $\tilde{R}$. Then, for every $0 < \Delta < 1$ there exists a discount factor $0 < \tilde{\gamma} < 1$ such that the MDP $\tilde{M}$ corresponding to $\tilde{R}$ and $\tilde{\gamma}$ is compatible to $M$.*

*Proof.* This is a special case of theorem 4.1. ∎

**Corollary A.1.1.** *In this setting, $\tilde{M}$ is compatible to $M$, if $\tilde{\gamma} > \Delta^{1/(w-1)}$.*

*Proof.* Assume that the agent is in an area where it can collect the reward $\Delta^k$. The optimal agent has to decide if it advances to the next "step" (where it can sustainably collect the reward $\Delta^{k-1}$) or if it stays in lower-reward areas. The return of the former is at least $\Delta^{k-1} \tilde{\gamma}^{w-1} / (1 - \tilde{\gamma})$, the return of the latter is at most $\Delta^k / (1 - \tilde{\gamma})$. Thus, the optimal agent will decide to steer towards the next step as long as $\tilde{\gamma} > \Delta^{1/(w-1)}$. ∎

Comparing this to corollary 4.1.1, we see that if $w < t_0$, this relaxes the lower bound on $\tilde{\gamma}$. A notable special case is $w = 1$, in which there exists a policy so that the reward is increased at every step. In this case, any $\tilde{\gamma} > 0$ is sufficient so ensure compatibility.

For reward functions that do not strictly adhere to definition A.1 but can be followed in a step-wise manner, the relation stated in theorem A.1 still qualitatively illustrates the following: The smaller the number of steps after which the agent can advance to a region of higher reward (i.e. the step width $w$), the more relaxed the lower bound on $\tilde{\gamma}$ becomes. In the important case that this step width becomes 1, this lower bound is 0.

