# OpenReview forum: "Plan-Based Relaxed Reward Shaping for Goal-Directed Tasks"
_ICLR.cc/2021/Conference — ICLR 2021 Poster_

### Official Review · AnonReviewer1 · 2020-10-27
**Review of Plan-Based Asymptotically Equivalent Reward Shaping**

**Rating:** 3
**Confidence:** 4

**Review:**

In this paper the authors introduce an asymptotic-equivalent (ASEQ) reward shaping reinforcement learning algorithm, which, they argue, is a relaxation of potential based reward shaping (PB). The authors present results suggesting that the proposed ASEQ method is significantly better than PB.

The authors state that this new method is a superset of PB (or a “subclass of reward shaping broader than PB”) which allows for a policy to be determined by shaping rewards on all intermediate states in a trajectory. There is unconvincing details on how this is done (eg. the authors state that it is ‘more direct’).

The authors make claims that this new method is suitable in difficult exploration problems in high-dimensional states spaces. They do not show evidence to support this claim and evaluate their method in a 10-d state space robotic manipulation task.

The contributions of this work are stated unconvincingly. For example, claims 1 and 2 are the same. The authors do not compare this method with other learning from demonstration methods (e.g. behavioural cloning or Generative Adversarial Imitation Learning)

Definition 4.1 is unclear and \mathcal{G} is undefined. There are assumptions that the authors state about the applicability of this method, such as at the end of section 4.1 regarding the likelihood of an agent leaving a well-defined volume of state space. The authors make claims that their method is asymptotically equivalent, that is in the long run, behaviour in a modified MDP will have the same result in a modified MDP. I would urge the authors to clarify this point.

The metic state-space (\mathcal{S}, d) is not well defined in Theorem 4.2.

The authors present their method as a modification to DDPG, and show that reward shaping leads to better performance on their tasks than no reward shaping. This is a likely outcome, given that they are providing the method with additional information.

I would urge the authors to compare against algorithms which use the same information, and stronger baselines rather than just ablative baselines on their method.  Finally, I do not think that this work is a good fit for ICLR given that it is more related to methods in reward shaping for reinforcement learning as opposed to the topics of more general greater interest to the audience at ICLR.

---

> ### Author Response · Authors · 2020-11-16
> **Thank you for the thorough feedback! (Part 1/2)**
>
> Q1. Concern on “why our proposed method is ‘more direct’ / effective than PB-RS”:
>
> We refer to our method as being “more direct” in the introduction and in the discussion. This is done in an attempt so summarize the explanation we give in section 4, and in particular in section 5.1, on how our method “asymptotically equivalent reward shaping” (ASEQ-RS) is different from potential-based reward shaping (PB-RS), and how this difference affects the training process in reinforcement learning. Roughly summarized, our proposed method is “more direct” in the sense that our formulation allows effective use of intermediate rewards in contrast to relying mostly on the discounted final reward.
>
> We’d be glad to change the wording if this causes this ambiguity, or else can you elaborate on which details you found unconvincing?
>
>
> Q2. Concern on “dimensionality of the problems we evaluated”:
>
> We refer to high-dimensional problems as a general motivation in the abstract and introduction, since sample efficiency becomes even more critical for such domains. In continuous control tasks, 10-D problems already pose challenges especially for long-horizon problems. This is verified by our analysis on a state-of-the-art RL method which fails to find a policy neither in no-reward shaping nor in PB-RS cases robustly. In particular, our claim is that the proposed method improves on the efficiency of PB-RS as the baseline method, which we show in our experiments.
>
>
> Q3. Concern on “convincing contribution statements”:
>
> In our view, contributions (1) and (2) differ in that contribution (1) states the introduction of our reward shaping technique (ASEQ-RS) itself, which is theoretically different than standard reward-shaping formulations, and (2) states that we apply this reward shaping technique to a plan-based setting, which is especially applicable and useful for robotic problems. ASEQ-RS can be applied in other settings as well, it is not limited to a plan-based setting. We updated our wording in the statement of contribution, hoping to make the distinction more clear.
>
>
> Q4. Concern on “comparison to learning from demonstration (LfD) methods”:
>
> Since our proposed method falls into an established line of work in reward-shaping, we preferred to compare our method to a state-of-the-art reward-shaping method (potential-based reward shaping). This has given us the chance to focus on the comparative advantages of using our formulation for the problem domains we are interested in, i.e., robotic manipulation tasks. We would also like to emphasize that our method is orthogonal to “methods in LfD that also exploit task information provided in the form of an approximate demonstration or plan”, as we articulate in the discussion, meaning that our method could even potentially be combined with some of the established approaches in LfD, which is definitely an interesting outlook for our future work.
>
>
> Q5. Concern on “undefined G in Definition 4.1”:
>
> In the original submission we write [notation was updated since]: “Let M be an MDP with state-space S. We call M convergent to G_a under optimal control iff there exists a set G_a ⊆ S such that [...]”. As we define in Def. 4.2 in the original submission (now also in Def. 4.1), G ⊆ S is the intersection of all such volumes G_a. If a more elaborate description is required, we’d be happy to expand on this definition and like to hear your feedback.
>
>
> Q6. Concern on “clarify claims that the method is asymptotically equivalent”:
>
> Thank you for this critical remark! We agree with your description (“same result in a modified MDP”) for the intuitive meaning of our definition of “asymptotic equivalence”. We formally introduce this definition in Def. 4.1 - 4.3 (now 4.1 -4.2). We then state theorems (4.1. and 4.2.) on how to construct reward functions (and MDPs) that fulfill this definition. We prove both these theorems. Is your comment referring to these proofs, and if so, could you elaborate on which part of the proof needs clarification?
>
> In the updated version we changed our wording from “asymptotically equivalent” to “compatible”, as this phrasing was imprecise and misleading for understanding the definitions. This, however, does not affect the mathematical definition and its resulting interpretation.
>
>
> Q7. Concern on “metric state-space is not well defined in Theorem 4.2”:
>
> Even though we are not sure if we understand this remark correctly, we try to provide an explanation. We’d be happy to provide additional clarification if we misunderstood your feedback.
>
> In essence, in order for theorem 4.2 to be applicable, we require that S is a metric space with metric d. We do not apply other constraints to d other than it being a metric. We made this explicit in the revised version of the paper by replacing “metric state-space(S,d)” --> “metric state-space(S,d) with metric d: S x S -> R”.

---

> > ### Author Response · Authors · 2020-11-16
> > **Thank you for the thorough feedback! (Part 2/2)**
> >
> > Q8. Concern on “baseline comparisons”:
> >
> > If we are interpreting this remark correctly, we believe our explanations might have led to a misunderstanding of our claims. Our ultimate goal is not a comparison of our method “asymptotically equivalent reward shaping” (ASEQ-RS) to no reward shaping (No-RS). We agree, this would be a trivial experiment, since indeed ASEQ-RS provides the agent with more information than No-RS. We only provide the result of No-RS for completeness. We do, however, compare our method (ASEQ-RS) against potential-based reward shaping (PB-RS), which is commonly used in the literature, and methodologically the closest to ours. PB-RS also provides the agent with reward shaping information. Therefore, PB-RS provides a fair baseline in our view, and we think that the observation we make, namely that ASEQ-RS significantly outperforms PB-RS using the same information in our experiments, is not trivially expected.
> >
> >
> > Q9. Concern on not being “a good fit for ICLR”:
> >
> > Since reinforcement learning is one of the key topics for ICLR, we believe any improvement of and/or augmentation on such methods is also a good candidate for the conference and its audience. More specifically, our method relies on an approximate planner, which works on an approximate representation of the real world. In that sense, our method exploits representations in order to improve the efficiency of RL in such settings. We hope that the explanations we provided here and the revisions on the paper would be helpful to clarify ambiguous points and emphasize this fitness better.

---

> ### Comment · AnonReviewer3 · 2020-11-23
> **Domain complexity**
>
> I would argue that the robotic manipulation task used in this work is sufficiently complex to demonstrate the effectiveness of the proposed shaping reward.  While there are certainly benchmark tasks with much higher-dimensional observation spaces (e.g. visual tasks like Atari), it can be argued that the states of many these environments can be captured by a much lower dimensional representations.  The exploration problem, that is, efficient coverage of the state space, is likely no more complex in these extremely high-dimensional domains than it is in lower dimensional control tasks.  The challenge in these higher dimensional domains may be better described as learning a good feature representation using only a sparse reward signal.  It would certainly be interesting to see whether the proposed reward shaping mechanism addresses this problem as well, but this is not essential to demonstrate that the method can be useful.

---

### Official Review · AnonReviewer2 · 2020-10-28
**A potentially useful approach to reward shaping in robotics settings.**

**Rating:** 7
**Confidence:** 4

**Review:**

POST-REBUTTAL UPDATE
======================

The rebuttal and resulting changes have addressed most of my concerns, and expanding the alternative reward-shaping aspect of the evaluation has helped too. I've increased the score.

======================


SUMMARY

The paper introduces a new, plan-based reward shaping approach to RL called ASEQ-RS. It is a generalization of potential-based reward shaping, one that doesn't have the same optimality guarantees but empirically facilitates faster policy learning.


HIGH-LEVEL ASSESSMENT

The paper's approach looks interesting, especially empirically. However, I have a number of concerns about the paper's theory: much of it seems to have errors. The paper can't be published with these errors, which is why I'm giving a slightly below borderline score for now. However, these error might be easily fixable. If the authors demonstrate a way to fix them without detracting much from the generality of the results, I'll increase the score, because the approach looks useful in practice.


TECHNICAL ISSUES/QUESTIONS

-- Def. 4.3 seems to have an error, in the sense that it doesn't quite have the interpretation stated in the paragraph below it. If an MDP has G = \emptyset (which is possible if different optimal policies lead to disjoint parts of the state space), it is asymptotically equivalent to any other MDP ver the same state and action space according to Def. 4.3, but it certainly doesn't mean that optimally behaving in this MDP and some other that has G \neq \empyset will have the same result. This may be fixable by excluding G = \emptyset MDPs from consideration, although I haven't fully verified this.

-- Def. 4.3 is also broken in the sense that the notion of equivalent it introduces is asymmetric, which contradicts the intuitive meaning of equivalence. Perhaps the easiest way to fix this one is to replace the term "equivalent" in this definition with something else.

-- Theorem 4.1's statement is ambiguous: when you say "Let M be convergent with optimal asymptotic volume G \subseteq B", are you assuming that M's asymptotic volume is always a subset of B? Or are you implicitly saying "*if* M's asymptotic volume is a subset of B, then let's call it G"? If it's the former, then that assumption is invalid -- due to \gamma being strictly less than 1, the optimal policy may eventually choose an action that lead to B with some probability and with some probability leads to a region with reward 0 (outside of B).

This can be quite an issue for Cor. 4.1.1 and Theorem 4.2, which are based on 4.1, and also in practice, because in non-ergodic MDPs with a discount factor the former version of the assumption can easily fail to hold, potenially breaking the paper's main results.

-- Theorem 4.1 loses its meaning if B = emptyset, in a similar way as Def 4.3.

-- The last paragraph on p.4 promises that the bound on gamma can be relaxed by Theorem 4.2, but I can't find a formal result that shows what the new bound on \gamma as a result of Theorem 4.2 is. This is important because in practice due to issues caused by function approximation, in RL one wants to set \gamma as low as possible. So what are the implications of Theorem 4.2 for choosing gamma?


Language errors:

"worst-case bound to the reward function:" --> "worst-case bound on the reward function:"

"lower bound to \gamma" --> "lower bound on \gamma"

"upper bound to" --> "upper bound on"

(Please search through the paper and fix all mistakes like the above.)

---

> ### Author Response · Authors · 2020-11-16
> **Thank you for the thorough feedback!**
>
> Q1. Concern on “interpretation of definition 4.3 if G can be empty”:
>
> Thank you for your remark; empty sets should be excluded, yes. In the updated version of the paper, we excluded empty sets from Def. 4.2. (now Def 4.1.) (optimal asymptotic, now called ‘optimal final’, volume) already. With this updated definition, if an MDP has an optimal final volume, this volume is non-empty per definition. As you suggested, this restores the interpretation of Def. 4.3 (now Def. 4.2) (asymptotically equivalent, now called ‘compatible’, MDPs) as stated in the text below.
>
>
> Q2. Concern on “use of the term ‘equivalent’ in definition 4.3”:
>
> We agree, using “equivalent” here is imprecise and misleading. We therefore updated our wording in the revised version of the paper as follows:
> 1. “optimal asymptotic volume” --> “optimal final volume”
> 2. “MDP \tilde{M} is asymptotically equivalent to M” --> “MDP \tilde{M} is compatible with M”
> 3. “Asymptotically Equivalent Reward Shaping (ASEQ-RS)” --> “Compatibility-Preserving Reward Shaping (CP-RS)
>
>
> Q3. Concern on “ambiguity of Theorem 4.1”:
>
> That’s a critical point that requires more clarity as you suggested. We meant to say the latter. Theorem 4.1 is supposed to be applicable for MDPs M with optimal asymptotic (now called optimal final) volume G, which in addition fulfils the requirement that G ⊆ B. We updated the wording accordingly to: “Let M be convergent with optimal asymptotic (now called optimal final) volume G, and let G ⊆ B”. In other words, M has to be such that the optimal policy is to keep the state within B after some finite time, and M has to be such that its dynamics allow for this (taking our pushing example, the optimal policy is to keep the box within the goal radius after finite time, which can be achieved by not touching it anymore).
>
>
> Q4. Concern on “excluding B = emptyset in Theorem. 4.1”:
>
> Thank you! We fixed this by excluding empty sets B in Theorem 4.1 and Theorem 4.2 in order to restore the meaning we intended while ensuring that downstream inferences remain valid.
>
>
> Q5. Concern on “choosing the value of gamma”:
>
> The last paragraph on p.4 was intended to refer to the appendix A4 (now appendix A5), where the formal result for the relaxed bound on gamma is stated. To make this clearer and self-contained, we appended the last paragraph on p.4 with a description of the results and implications of appendix A5 for choosing gamma, so that the reader does not need to skip over. For the full derivation, we are forced to refer the reader to the appendix due to space restrictions, unfortunately.
>
> Roughly summarized, the implications following from the theorem in appendix A5 for choosing gamma are that, if a reward function can be followed in a stepwise fashion with step width w, the lower bound in on gamma is as in Corollary 4.1.1., but replacing t_0 with w. Notably, this means that if the reward function can be followed monotonically (w=1), any choice of gamma>0 is possible.
>
>
> Q6. "Language errors":
>
> Thank you for those fixes. We applied them throughout the paper.

---

> > ### Comment · AnonReviewer2 · 2020-11-19
> > **Thanks, this addresses my concerns.**
> >
> > I've updated the review and increased the score.

---

### Official Review · AnonReviewer3 · 2020-10-28
**This work presents asymptotically-equivalent reward shaping as an alternative to potential-based shaping.  Theoretical results show that for certain goal-oriented MDPs, the much more flexible class of ASEQ shaping rewards yield policies that converge to the same goal regions as optimal policies for the original MDP.**

**Rating:** 7
**Confidence:** 4

**Review:**

SUMMARY OF CONTRIBUTION:

This work presents asymptotically-equivalent (ASEQ) reward shaping as an alternative to potential-based shaping.  ASEQ shaping admits a more general class of augmented reward functions than potential based shaping, at the cost of a weaker notion of equivalence between the classes of optimal policies for the original MDP and the augmented MDP.  ASEQ assumes that the original MDP uses a binary reward function that is 1 for all states in a subset of the state space to which any optimal policy will converge (effectively the "goal" of the MDP), and 0 elsewhere.  The theoretical results presented in this work show that for any augmented reward function which is between 0 and 1 outside of the goal region, and for a sufficiently large discount factor, the an optimal policy for the augmented reward will converge to this goal region.  The flexibility of this form of reward shaping with a plan-based shaping reward that provides smaller rewards for reaching waypoints along a path to the goal.  The advantage of this form of shaping reward over more limited potential-based shaping is demonstrated on a set of simulated robotic manipulation tasks.

AREAS OF CONCERN:

One potential issue with the experimental results is that the potential function used for the baseline is defined identically to the plan-based shaping reward used with ASEQ shaping.  While this is the most direct comparison between the two approaches, it is possible that using potential-based shaping with a simpler potential function, such as the distance between the box and the goal, would lead to better performance for potential-based shaping.

As reward shaping is a meta-algorithm that can be applied to almost any RL method, it would also be useful to evaluate ASEQ shaping on multiple RL algorithms, not simply DDPG, to show that the benefits of the more flexible class of shaping rewards are not limited to a single class of algorithms.

There are some places where the notation could be improved:
  1) The notation in Equations (1) and (2) is somewhat awkward, as q_pi appears to be a distribution over transitions in (1), but is defined as a distribution over entire trajectories in (2).
  2) It isn't entirely clear what the subscript 'a' of the sets G_a in definitions 4.1 and 4.2 refers to.

CONCLUSION:

While the form of shaping reward used in this work is not itself novel, the work does for the first time formally describes the class of problems to which this more general form of reward shaping can be applied, with conditions on the form of the shaping reward that guarantee compatibility with the original task.  The work does have its limitations, both in terms of the class of problems to which ASEQ shaping can be applied, and in terms of the depth of the empirical evaluations, but it will be of value to the community nonetheless.

---

> ### Author Response · Authors · 2020-11-16
> **Thank you for the thorough feedback!**
>
> Q1. Concern on “(potential-based) shaping function used for comparison”:
>
> Thank you for your remark. We agree that this discussion was missing and is important. We therefore appended section 5.2. with a discussion of additional preliminary experiments on the shaping function used for comparisons. These experiments are presented in detail in appendix 1 (moved to the appendix due to space restrictions). Here we compare different baseline choices for PB-RS, which can be independent of the shaping function for ASEQ-RS. We also followed your suggestion to use the distance between the box and the goal as a potential function. We find that indeed this is a very good choice for simpler problems, but it does not scale well to more complex settings, where the end effector has to change its contact point with the box multiple times, and navigate around it. We explain both our choices for the potential-based baseline and for the asymptotically equivalent reward shaping function in the revised paper. For further discussion of a similar concern, please also refer to Q3 and Q4 in our answer to AnonReviewer4.
>
>
> Q2. Concern on “use with other RL methods”:
>
> Thank you for your remark. We agree that it would be interesting to apply ASEQ-RS not only to DDPG, but also to other RL algorithms. The focus of the current work on manipulation experiments made DDPG a suitable choice. Having said that, we are already looking at applications of ASEQ-RS other than DDPG for future work.
>
>
> Q3. Concern on “some places where the notation could be improved”:
>
> Thank you for pointing out these improvements. We updated our notation accordingly. With regard to (2), we merged definitions 4.1. and 4.2. in order to make the relation between G and G_a clearer.

---

> ### Author Response · Authors · 2020-11-20
> **Update on Q2: Concern on “use with other RL methods”**
>
> Following your suggestion, we now applied our method to Proximal Policy Optimization (PPO) as well. We chose PPO since being both on-policy and stochastic, it differs from DDPG in two relevant ways. We now revised the paper again to discuss the results from these experiments. Due to space restrictions, we show detailed results in appendix 4.
>
> We find that the results from our initial experiments are confirmed in the experiments using PPO. The significant increase in sample efficiency when using our method over PB-RS is reproduced in a similar way.
>
> Thank you again for your helpful remark!

---

> ### Comment · AnonReviewer3 · 2020-11-23
> **Response to Rebuttal**
>
> I believe the authors have addresed my concerns in the updated version of the paper.  I have left my score unchanged, as it primarily relfects the work's overall potential for impact on the field.

---

### Official Review · AnonReviewer4 · 2020-10-28
**The ideas in the paper are potentially interesting, but the assumptions appear somewhat restrictive, the experiments are unconvincing, and there are a couple of technical details that lack discussion.**

**Rating:** 6
**Confidence:** 3

**Review:**

= Overview =

The paper proposes "Asymptotically equivalent reward shaping", a new reward shaping approach that extent potential-based reward shaping to allow for shaping functions that -- even if eventually altering the resulting optimal policy -- retain key elements in the performance of such policy, namely ensuring that the resulting policy leads the agent to a subset of the states that the original optimal policy does. The paper also proposes a method to compute a shaping function from behavior demonstration.

= Positive points =

The idea of reward shaping is important in reinforcement learning, as it can have an important impact in terms of sample complexity. As far as I know, this area of research has not seen significant advances or new ideas in recent years. The paper provides a fresh view of this problem, by proposing to alleviate the requirement that the shaped function should lead to the same optimal policies.

= Negative points =

The proposed approach seems to be restricted to a very narrow class of MDPs. Additionally, the proposed shaping function is presented with little motivation, intuition or discussion whatsoever. Finally, the experimental evaluation is not very convincing, as the proposed approach is compared with "arbitrary" potential based shaping functions (more details below).

= Comments =

The paper addresses a problem that, in my view, is relevant to the RL community and has not seen many advances in recent years. Its premise is that the "classical" reward shaping may be too restrictive, in that it requires the optimal policies before and after shaping should be the same. The paper instead proposes to alleviate this requirement, as long as the resulting policies lead to "similar" long-term behavior.

This notion is, in my view, interesting and in line with work on intrinsic motivation, by Barto and collaborators, and in reward optimization conducted by Sorg, Sequeira and others. In this respect, it could be relevant to at least mention this line of work in the paper.

In addressing reward shaping, however, the paper restricts to problems where the reward is binary (i.e., 0 most of the time, and 1 in some states). These are, essentially, goal-driven MDPs, where the optimal policy just drives the agent towards one of the states with non-zero reward. The paper notes that, if the reward is shaped by providing sufficiently small reward values in the non-goal states and the discount is made sufficiently large, the optimal policy will still lead the agent to one of the goal states. This is a relatively straightforward conclusion, and the whole shaping strategy proposed by the paper follows from this simple observation.

The paper then proposes a shaping function from a given demonstration (consisting of a sequence of states). Unfortunately, the presented shaping function is introduced with no motivation, intuition, or discussion whatsoever, and one wonders why such shaping function is a good idea.

The paper concludes with an experimental evaluation section, where the proposed approach is compared against potential-based reward shaping and shown to outperform the latter. However, the potential-based shaping functions used for comparison are introduced in an ad hoc manner, and it is not clear to me why these shaping functions are good baselines for comparison.

Finally, on a small note, I feel that the paper could benefit from cleaning up/streamlining the notation and language. For example, why the proposed notion of "asymptotic equivalence" is called asymptotic of equivalence is unclear, since the notion of "optimal asymptotic volume", on which the former builds, is defined for a finite time (not asymptotic), and the relation does not seem to be an equivalence (if for no other reason, it does not seem symmetric to me). Definitions 4.1-4.3 make use of heavy notation but translate a very simple notion: that an MDP M is "asymptotically equivalent" to an MDP M' if the optimal policy of the former leads the agent to a subset of the goal states of the latter.

---

> ### Author Response · Authors · 2020-11-16
> **Thank you for the thorough feedback! (Part 1/2)**
>
> Q1. Concern on ”related work on intrinsic motivation and reward optimization”:
>
> Thank you for pointing this out. We agree that our work is related in spirit to prior work on intrinsic motivation and reward optimization. Both lines of work offer alternative solutions to the sparse-reward problem outside of potential-based reward shaping. We updated our related-work section to point this out.
>
>
> Q2. Concern on “restricted space of problems / goal-driven MDPs”:
>
> It is true that our work is restricted to goal-driven MDPs, and we updated the discussion to emphasize this point more carefully. However, we would argue that (1) this class of MDPs is quite common and important for many applications and (2) applying reinforcement learning in such MDPs still poses a big challenge. We specifically have robotic manipulation tasks in mind, which are very often goal-driven. A typical example here would be to place an object at a certain position using a tool, while intermediate rewards are often not informative. Such sparse reward settings still pose a major challenge to reinforcement learning, which is where we aim to contribute with the presented shaping method.
>
>
> Q3 and Q4. Concern on “motivation for the shaping function” and on “the potential-based shaping function used for comparison as a baseline”:
>
> We admit that these are important points that were missing. We would like to address them together, since they both concern the choice of shaping functions for PB-RS and for ASEQ-RS.
>
> Intuitively, the Gaussian shaping function we used was chosen since it rewards moving closer towards the plan and moving further along the plan in a straightforward way. It is similar to the potential function used in the work by Brys et al. (2015), which is concerned with a comparable setting, but uses PB-RS. The exact shaping function we used was then found in preliminary experiments. However, we had not yet reported on (1) why we chose this particular shaping function for ASEQ-RS and (2) whether it was “fair” to choose the same shaping function as a potential function for the PB-RS baseline.
>
> To answer both these questions, we appended section 5.2. with a discussion that also references additional preliminary experiments we included now. These experiments are presented in detail in appendix 1 (moved to the appendix due to space restrictions). Here we analyze the use of different shaping functions both for ASEQ-RS and PB-RS. We look at both gaussian functions with varied width parameters and at a simpler distance function as suggested by AnonReviewer3. We vary the shaping functions for ASEQ-RS and PB-RS independently from each other, analyzing our choice both for (1) and (2). The brief summary of our findings is as follows:
> 1. Our findings confirm that our initial choices of potential functions both for ASEQ-RS and PB-RS allow for a fair comparison, in the sense that this choice does not give an unfair advantage to our method (ASEQ-RS) over PB-RS.
> 2. However, we were able to increase the performance of both methods by choosing a different Gaussian width parameter as a result of this analysis. This increase in performance affects both ASEQ-RS and PB-RS in a similar way, and therefore does not change the relative advantage of ASEQ-RS over PB-RS that we initially found.
> We repeated all experiments shown in the paper with the updated shaping functions. We find that the qualitative findings of our initial submission remain unchanged. We updated the experimental data in the revised paper accordingly.
> 3. We repeated all experiments shown in the paper with the updated shaping functions. We find that the qualitative findings of our initial submission remain unchanged. We updated the experimental data in the revised paper accordingly.
>
>
> Q5. Concern on “calling our method asymptotically equivalent”:
>
> Thank you for these suggestions! We agree that our wording here is imprecise and misleading, both in terms of using “asymptotic” and in terms of using “equivalent”. While we would like to mention that the stated theoretical introduction of our method is independent of this wording and its “meaning” is unchanged, however it does potentially convey a wrong understanding of our method to the reader. We therefore updated our wording in the revised version of the paper as follows:
> 1. “optimal asymptotic volume” --> “optimal final volume”
> 2. “MDP \tilde{M} is asymptotically equivalent to M” --> “MDP \tilde{M} is compatible with M”
> 3. “Asymptotically Equivalent Reward Shaping (ASEQ-RS)” --> “Compatibility-Preserving Reward Shaping (CP-RS)

---

> > ### Author Response · Authors · 2020-11-16
> > **Thank you for the thorough feedback! (Part 2/2)**
> >
> > Q6. Concern on “Definitions 4.1-4.3 [now 4.1-4.2] make use of heavy notation but translate a very simple notion”:
> >
> > We agree that the notion of "asymptotically equivalent" is rather simple and intuitively understood. The aim of Definitions 4.1-4.3 (now 4.1-4.2) is to provide a theoretical grounding, or “language”, for this type of relation between MDPs, and making this easy for the reader to follow. We therefore followed your suggestion and streamlined the notation in the updated version of our paper, e.g., by merging definition 4.1 and 4.2, and updating our wording from “asymptotically equivalent” to “compatible”.

---

> > > ### Comment · AnonReviewer4 · 2020-11-21
> > > **Response to authors**
> > >
> > > I thank the authors for their response. They addressed my comments satisfactorily and, as such, I improved my recommendation - although I am still somewhat setback by the limitation to goal-driven problems.

---

### Author Response · Authors · 2020-11-16
**Paper update overview**

We would like to thank the reviewers for their constructive and insightful feedback. We tried to address their concerns and revised the paper accordingly, and will also answer directly to their reviews. These are the most notable changes:

- We fixed errors in our definitions in theorems without breaking the meaning and implications we claimed in the initial submission.
- We included a discussion and motivation of our choice of shaping function used for ASEQ-RS. This discussion also relies on an additional experimental study we included.
- Similarly, we discussed and motivated our choice of potential function for the PB-RS baseline, and conducted experiments to support this discussion as well.
- As a result of this, we also updated all other experiments. The qualitative findings remain unchanged.
- We addressed remarks regarding the applicability of our method to different types of MDPs in the discussion.
- We updated the related-work section to also discuss work in the areas of intrinsic motivation and reward optimization.
- We changed our wording from “asymptotically equivalent” (ASEQ) to “compatibility-preserving” (CP).
- We streamlined the notation at several places in the paper, and fixed several ambiguities and language errors in our presentation.

We are happy to provide further clarification if any of the reviewers’ concerns are not answered.

---

### Author Response · Authors · 2020-11-20
**Second paper update overview**

We revised the paper again to introduce the following updates:

- The comparisons of our method to the potential-based baseline shown in the initial version were obtained using Deep Deterministic Policy Gradient (DDPG). We now conducted additional robotic manipulation experiments using Proximal Policy Optimization (PPO) as well. In contrast to DDPG, PPO is on-policy and stochastic. The experiments using PPO confirm the qualitative findings from the experiments using DDPG; we find that our method significantly increases the sample efficiency over PB-RS here as well. A detailed description can be found in appendix 4 of the revised paper. We reference these experiments in the main script.
- In the first revision, we changed our wording from “asymptotically equivalent” (ASEQ) to “compatibility-preserving” (CP). We now decided to instead change our wording from “asymptotically equivalent” (ASEQ) to “final-volume-preserving” (FV). We hope that this updated wording is more descriptive of our method.

We are happy to provide further clarification if any of the reviewers’ concerns are not answered.

---

> ### Comment · AnonReviewer2 · 2020-11-22
> **Revise the title?**
>
> This doesn't affect my assessment of this work's merits, but I'd  suggest revising the current title: the "final-volume-preserving" part is quite cryptic. Perhaps "Plan-based reward shaping for goal-directed tasks"? Many options here; it's fine for the title to be not as precise as it currently is, but it would be good to make something that readers can immediately understand and relate to.

---

> > ### Author Response · Authors · 2020-11-23
> > **Re: Revise the title?**
> >
> > We tried to make the title as descriptive as possible, but we agree that it probably has become too cryptic now. We like your suggestion for the title; we would just like to add a slight modification: Since “reward shaping” is sometimes used as a synonym for “potential-based reward shaping”, we would like to emphasize in the title already that our method in fact differs from potential-based reward shaping. Therefore, we would suggest using “Plan-based _relaxed_ reward shaping for goal-directed tasks” as the title. We will update the paper accordingly. Thank you for your remark!

---

### Author Response · Authors · 2020-11-23
**Updated Title**

We now updated the title to "Plan-Based Relaxed Reward Shaping for Goal-Directed Tasks". It was suggested that although being descriptive of our method, the former title "Plan-Based Final-Volume-Preserving Reward Shaping" was not easy to understand. Thank you for pointing this out!

---

### Decision · Program_Chairs · 2021-01-07
**Final Decision**

**Decision:**

Accept (Poster)

**Comment:**

Summary:
This paper introduces a novel type of reward shaping (not potential-based) that can work for MDPs with 0/1 rewards (e.g., systems with goal states). I think the paper contains a nice nice of empirical results, clear explanations/insights, and theoretical contributions.
As long as the authors do a good job saying up-front what kinds of MDPs the method does/does not address, this would provide an interesting addition to practical methods for solving sparse-reward RL tasks.
The paper could be strengthened with additional empirical results, but that is almost always true, and I think the number and quality of experiments are well above the bar.
The paper could also be strengthened if it better compared with other ways of using demonstrations. This is done in the text, but not empirically, as the authors wanted to focus on reward shaping methods. I think this is a relatively minor point and does not outweigh the many positives.

Discussion:
One reviewer remains against accepting this paper. I respectfully disagree with their evaluation and unfortunately they did not update their review after the responses. Hopefully they would agree the final version of this paper is indeed a useful contribution.

Recommendation:
I believe this paper should be accepted based on the science.